# Different Species of Marine Sponges Diverge in Osteogenic Potential When Therapeutically Applied as Natural Scaffolds for Bone Regeneration in Rats

**DOI:** 10.3390/jfb14030122

**Published:** 2023-02-24

**Authors:** Cíntia P. G. Santos, João P. S. Prado, Kelly R. Fernandes, Hueliton W. Kido, Bianca P. Dorileo, Julia R. Parisi, Jonas A. Silva, Matheus A. Cruz, Márcio R. Custódio, Ana C. M. Rennó, Renata N. Granito

**Affiliations:** 1Department of Biosciences, Federal University of São Paulo (UNIFESP), Santos 11015-020, SP, Brazil; 2Department of Physiotherapy, Federal University of São Carlos (UFSCar), São Carlos 13565-905, SP, Brazil; 3Institute of Biosciences, University of São Paulo (USP), São Paulo 05508-000, SP, Brazil

**Keywords:** bone tissue engineering, demospongiae, natural biomaterial, marine sponges, scaffolds

## Abstract

A highly porous structure, and an inorganic (biosilica) and collagen-like organic content (spongin) makes marine sponges potential candidates to be used as natural scaffolds in bone tissue engineering. The aim of this study was to characterize (through SEM, FTIR, EDS, XRD, pH, mass degradation and porosity tests) scaffolds produced from two species of marine sponges, *Dragmacidon reticulatum* (DR) and *Amphimedon viridis* (AV), and to evaluate the osteogenic potential of these scaffolds by using a bone defect model in rats. First, it was shown that the same chemical composition and porosity (84 ± 5% for DR and 90 ± 2% for AV) occurs among scaffolds from the two species. Higher material degradation was observed in the scaffolds of the DR group, with a greater loss of organic matter after incubation. Later, scaffolds from both species were surgically introduced in rat tibial defects, and histopathological analysis after 15 days showed the presence of neo-formed bone and osteoid tissue within the bone defect in DR, always around the silica spicules. In turn, AV exhibited a fibrous capsule around the lesion (19.9 ± 17.1%), no formation of bone tissue and only a small amount of osteoid tissue. The results showed that scaffolds manufactured from *Dragmacidon reticulatum* presented a more suitable structure for stimulation of osteoid tissue formation when compared to *Amphimedon viridis* marine sponge species.

## 1. Introduction

Biomaterial-based therapy has increasingly become a viable strategy for treating bone fractures. Synthetic or naturally occurring, every type of biomaterial has a unique set of characteristics. Polyurethanes, polyesters, metals such as titanium and other synthetic polymers have advantageous properties over natural ones especially because of their abundance, limitless designs and customizable attributes. They require, however, chemical alterations since they lack cell adhesion sites and are, therefore, considered to be less biocompatible [1,2].

As a vast source of natural biomaterials, marine biodiversity has lately been gaining prominence in scientific research around the world. Among the invertebrates that stand out in different studies, marine sponges (Phylum Porifera) are primitive sessile animals characterized by being multicellular, filter-feeding and structurally porous [3].

It is indeed a fact that these animals archaically filter water for feeding with microorganisms that confer them unique characteristics for their use as biomaterials in the engineering of bone tissue. Their porous architecture with interconnected pores would favor, once implanted in patients, the passage of cells and blood vessels and, consequently, tissue regeneration [3]. Thus, adequate structural characteristics of the implant play a fundamental role in the efficiency of the treatment since the formation of the new tissue depends on an adequate migration and proliferation of the cells responsible for this process and also on the invasion of new blood vessels within the implanted scaffold. This infiltration will only properly occur if the scaffold is structurally favorable, especially in terms of porosity, pore size and pore interconnectivity. In addition to being a source of precursor cells that will be later responsible for the new tissue formation, vascularization may also have its relevance since it provides the nutrients needed for cellular metabolism and the removal of its residual products [4,5].

Besides their structural peculiarities, marine sponges have, like the bone itself, a skeleton composed of an organic component, named spongin, and inorganic components such as biosilica [6,7]. Spongin is a protein similar to vertebrate collagen that has also been used in the production of scaffolds for tissue bioregeneration [8] because it allows cell attachment, proliferation and migration through the biomaterial [9]. Based on this, spongin is an excellent alternative to animal collagen, due to the low risk of transmission of infection-causing agents and good biocompatibility [10,11]. Biosilica is an inorganic element that is known to drive bone cell differentiation and increase mineralization [6,11,12].

The ideal scaffold for bone substitution, regardless of whether it is natural or synthetic, needs to be biocompatible, porous, have osteoconductive and osteoinductive capacities, mechanical properties similar to those of bone and a rate of degradation compatible with the process of bone remodeling [3,13].

Previous in vitro studies showed that osteoprogenitor cells were able to grow and attach onto the sponge skeleton. Green et al. [14] demonstrated that the skeleton of *Spongia* sp. enabled human osteoprogenitor cells’ adhesion, expansion and invasion. Alkaline phosphatase and type I collagen histochemical stains showed that the bone matrix could be formed. Lin et al. [15], by also evaluating a sponge skeleton with a collagenous fibrous network (*Callyspongiidae* sp.), showed that mouse primary osteoblasts were able to anchor onto the surface of collagen fibers, express osteoblast markers (osteocalcin and osteopontin) and form mineralization nodules. Likewise, SaOS-2 cells were able to grow and colonize the bioceramic structure of *Petrosia ficiformis* sponges after calcination [16], addionally demonstrating the advantages conferred by sponge scaffold architecture as a template for bone cell growth, differentiation and mineralization [17].

These ancient multicellular organisms boast a diverse array of skeletal structures that have evolutionary-approved 3D-scaffold-like qualities and appear to be highly suitable for use across a range of fields within modern bioinspired materials science, biomimetics and regenerative medicine [16]. Even though several in vitro studies speculate its potential for use as a scaffold, only one study has been conducted with the aim of analyzing the in vivo potential of using marine sponges in their natural state until now. Nandi et al. [18] carried out an investigation to identify and characterize marine sponges as potential bone scaffolds. For this purpose, samples of marine sponge *Biemna fortis* (class Demospongiae), collected from the intertidal region of Anjuna, India, were implanted in femoral defects in rabbits. After 90 days, the results showed that marine sponges of the mentioned species, combined or not with growth factors (IGF-1 and BMP-2), were biocompatible and biodegradable, and could be considered as a new natural biomaterial for bone tissue engineering purposes.

Based on the above, sea sponges, with their porosity, biosilica and spongin, can be potentially considered as innovative bone substitutes, which would allow the development of new therapeutic resources for the improvement of patients’ quality of life. Additionally, since they a natural, abundantly found material, their use would contribute to a reduction in the treatment costs in public health systems. However, this potential is still largely unexplored. Despite several studies evaluating the use of compounds extracted from marine sponges, only one in vivo study was found in the literature evaluating the marine sponge skeleton as a bone-mimicking biomaterial. Moreover, this single study did not perform quantitative evaluations of the bone healing process. Finally, since sponges structurally differ among species, we hypothesize that their osteogenic potential would also be different. In this way, the present study performed a physical-chemical characterization and an evaluation of the in vivo osteogenic potential of scaffolds manufactured from two species of marine sponges with the aim of developing efficient substitutes for guided bone regeneration. The species of Porifera used in this study were *Dragmacidon reticulatum* and *Amphimedon viridis*, which are abundantly present on the Brazilian coast.

## 2. Materials and Methods

It is important to note that this study is registered in the National Management System of Genetic Patrimony (Sistema Nacional de Gestão do Patrimônio Genético, SisGen, registration number A56D034).

The marine sponge species *Dragmacidon reticulatum* and *Amphimedon viridis*, both belonging to the class Demospongiae, were used in this study. These two species (Figure 1) were specifically chosen for being abundant and easily found on the Brazilian coast. Both have porous skeletons composed of spongin and biosilica, which are interesting elements in the context of bone repair. The sponges were collected in high hydrodynamic coasts, in the intertidal zone, in the area of São Sebastião, Brazil (23°49′23.76″ S, 45°25′01.79″ W) and in the area of Enseada do Araçá (23 No. 81′73.78″ S, 45° 40′66.39″ W, São Sebastião, Brazil). An amount of 200 g of each marine sponge species was collected for this study.

After the collection, the sponges were washed with running water, classified according to their species’ characteristics and kept in the freezer until use. To produce the scaffolds, the sponges were cut with a trephine-type dental drill (3i Implant Innovations Inc., Palm Beach Gardens, Florida, USA) and a scalpel blade for the manufacture of 3 mm diameter × 2 mm thick scaffolds (Figure 1). The produced scaffolds were freeze-dried and sterilized by ethylene oxide (Acecil Central de Esterilização Comércio e Indústria Ltd.a—Campinas/SP, Brazil). The parameters used for the freeze drying were −40 °C and 600 uHG (DIM Liofilizador LT X.X00—Terroni Equipamentos Cientificos Ltd.a—São Carlos/SP, Brazil).

Characterization analyses of the scaffolds were performed before and after an incubation period, the conditions of which were determined according to the Kokubo protocol [19,20,21]. For the SEM, FTIR, XRD and EDS aanlyses, the scaffolds were incubated in simulated body fluid (SBF; pH 7.4) in a ratio of 1:10 (mass of the material (g): volume of the SBF (mL)) during a period of 21 days. Samples were evaluated before (day zero) and after incubation at periods 1, 7 and 21 days. For pH and mass loss evaluations, the scaffolds were placed in 3 mL of phosphate-buffered saline (PBS, pH 7.4) and incubated at 37 °C in a water bath on a shaker table (70 rpm) for 21 days. These analyses were performed before (day zero) and after incubation of scaffolds at periods 1, 3, 7, 14 and 21 days.

### 2.1. Surface Morphology Analysis (SEM)

A scanning electron microscope (SEM, Le0 440, Carl Zeiss, Jena, Germany), operating at a 10 keV electron beam, was used to morphologically analyze the scaffolds surface. This technique consists of obtaining the enlarged image of the sample from the interaction of an electron beam with the material. For this, the samples were fixed on an aluminum base using a carbon tape. Next, due to the non-conductive properties of the samples (organic origin), they were covered with a thin layer of conductive material; in this case, the material used was gold. Thus, images were obtained with magnification of 500×, in the above-mentioned periods, in order to assess the morphology of the initial surface and the degradation behavior of the biomaterials (*n* = 5).

### 2.2. Fourier Transform Infrared Spectroscopy (FTIR)

To identify the chemical bonds present in the material, Fourier-transform infrared spectroscopy (IRAffinity-1S-FTIR Shimadzu spectrophotometer, São Paulo, Brazil) was used. The specters were obtained in the range of 4000–400 cm^−^^1^, with a resolution of 4 cm^1^ (*n* = 3).

### 2.3. X-ray Diffraction (XRD)

The crystalline phase of the material was evaluated by X-ray diffraction (XRD) with Philips X’Pert MPD diffractometer, Cu-Ka (l = 0.154 nm), 45 kV, 30 mA. Data were collected at angles between 20° and 60° at the IQSC(USP), São Paulo (*n* = 3).

### 2.4. Energy Dispersive X-ray Spectroscopy (EDS)

This analysis was carried out by means of equipment IXRF Systems 500 coupled to a scanning electron microscope (SEM) that allowed a qualitative evaluation of the chemical elements present in the samples, with 0.5% mass detection [22] (*n* = 5).

### 2.5. pH Evaluation

Directly after removing samples from the incubation medium (*n* = 5), pH of incubation medium was measured with a pH meter (Orion Star A211, Thermo Scientific, Waltham, MA, USA).

### 2.6. Degradation Analyses

The mass loss evaluation was performed to determine the degradation of biomaterials in liquid medium. After the experimental periods of incubation in PBS, the samples were oven-dried overnight at 37 °C and weighed on a precision balance. The relation between the final weight obtained and the initial weight was calculated to quantify the mass loss in percentage, according to the following equation: % Mass loss = ((fm − im)/im) × 100%, where fm is the sample mass after immersion in PBS and im is the sample mass before immersion in the same solution (*n* = 5).

### 2.7. Porosity Evaluation

The porosity of *Dragmacidon reticulatum* and *Amphimedon viridis* scaffolds were evaluated according to the Archimedes’ principle (Equation (1)). First, scaffolds (1 cm × 0.4 cm) were obtained by cutting the sponges with a cutter. The scaffolds were dried overnight at 37 °C. The scaffolds were then weighed on a precision balance. The density of water was also calculated (mass/volume). Later, the scaffolds were placed in a glass container with 5 mL of water and the values of mass (grams) and volume (mL) were recorded. The scaffolds were removed from the glass container and the weight of the water was again recorded. The scaffolds’ density was calculated using the values of water + scaffolds and water weight differences. For the porosity calculation, the following formula was used:Porosity (%) = m1 − m3.WD[(m1.WD) + (SM.SD)] − (m3/WD) × 100(1)
where m1 is initial mass of water, m3 is water mass after removing the scaffold, WD is the water density, SM is the scaffold mass and SD is the scaffold density (*n* = 5).

### 2.8. Biological Evaluation

#### 2.8.1. In Vivo Study

Thirty male Wistar rats were used (12 weeks old, weight 300–350 g) in this study. All rats were submitted to a surgical procedure in which a unicortical non-critical bone defect was performed in both tibias. The animals were randomly divided into 3 groups: control group (CG)—defects were left unfilled; *Dragmacidon reticulatum* group (DR)—defects were implanted with scaffolds of marine sponge *Dragmacidon reticulatum*; *Amphimedon viridis* group (AV)—scaffolds that were implanted belonged to the species *Amphimedon viridis*. After surgical procedure, the animals were kept at a controlled temperature, 12 h light–dark period, with free access to water and standard food. This study was approved by the Ethics Committee on the Use of Animals (CEUA) of the Federal University of São Paulo (2017/3011170417).

#### 2.8.2. Surgical Procedures

A non-critical-sized bone defect, 3 mm in diameter, was performed in the upper third of each rat tibia (10 mm below the knee joint) by using a motorized trephine drill (3i Implant Innovations Inc., Palm Beach Gardens, Florida, USA) irrigated with saline solution (Figure 2A). Then, the wounds were closed with resorbable Vicryl^®^ 5-0 (Johnson & Johnson, Sint-Stevens-Woluwe, Belgium). Surgeries were performed according to the ethical principles of animal instrumentation, at standard conditions of asepsis and general anesthesia. Initially, the animals were anesthetized with intraperitoneal injection of ketamine (80 mg/kg), xylazine (8 mg/kg), acepromazine (1 mg/kg) and fentanyl (0.05 mg/kg) in a single syringe. In addition, a single dose of cephalothin antibiotic (60 mg/kg) was given preoperatively. Next, trichotomy and antisepsis were performed with the aid of a shearing machine and sterile gauzes containing 2% degermant iodine, which was followed by three steps of 70% ethanol application in the surgical focus.

All animals were then submitted to the surgical creation of bone defects bilaterally in the tibias, but only the animals of DR and AV groups received implants (scaffolds) as treatment. After surgery, anti-inflammatory meloxicam was administered subcutaneously at a dose of 2 mg/kg and, after 24 and 48 h, at a dose of 1 mg/kg. Finally, the animals were placed in individual boxes with free access to water and food and were monitored until anesthesia was completely over. Additionally, postoperative animals were monitored daily throughout the whole treatment period, with pain parameters being constantly evaluated. The animals were euthanized by drug overdose (intraperitoneal injection of ketamine 240 mg/kg and xylazine 24 mg/kg) 15 days after surgery.

#### 2.8.3. Histological Procedures

After sample collection, the left tibias were dehydrated with 70% ethanol for three days. They were then dehydrated in absolute ethanol (100%) for a further three days, diaphanized in toluene for one day and included in the methyl methacrylate resin (Merck acrylic resin). The obtained blocks were sanded and cut using a microtome (Leica Microsystems SP 1600, Nussloch, Germany). Five-micrometer-thick sections were perpendicularly obtained, considering the medial–lateral axis of the implants. Histological sections were stained with Goldner’s tri-chromium (Merck).

#### 2.8.4. Histological Analysis

The qualitative analysis of the slides was performed by means of the morphological description of bone defects, according to the following criteria: presence of newly formed bone tissue (primary and secondary bone), granulation tissue, presence of fibrosis and biomaterial. Analyses were performed blindly on the 10× objective.

#### 2.8.5. Histomorphometric Analysis

A microscope (Labophot 2ª, Nikon, Minato City, Tokyo) coupled to the OsteoMeasure software (OsteoMetrics, Atlanta, GA, USA) was used for the quantitative analyses. Measurements were performed in the fields located in the medial region of the bone defect, from the upper border until the bottom of the defect, using the 10× objective. The total region of interest (ROI) was 1.85 ± 0.38 mm^2^ (Figure 2B). The following histomorphometric parameters were obtained: bone volume as percentage of tissue volume (BV/TV%), osteoid volume as percentage of tissue volume (OV/TV%), number of osteoblasts per unit area of tissue analyzed (N.Ob/T.Ar mm^2^), osteoblastic surface as a percentage of bone surface (Ob.S/BS%) and percentage of fibrous tissue volume as a percentage of tissue volume (Fb.V/TV%), according to international standardized nomenclature [23]. A parameter was additionally included in order to analyze the biomaterial present in the ROI: biomaterial volume as a percentage of tissue volume (Bm.V/TV%) (*n =* 5). Active (mature) bone-forming osteoblasts were identified by their cylinder-like shape and their arrangement in rows over an area of osteoid tissue.

#### 2.8.6. Biomechanical Test

Biomechanical analysis was performed using the three-point bending test on the right tibia of animals of all groups. Biomechanical assays were performed on the Instron universal testing machine (model 4444, 825 University Ave Norwood, MA, 02062-2643, US) at room temperature. For the test, a load cell with a maximum capacity of 1 N and a preload of 5 N and a constant speed of 0.5 cm/min was used. Both tibia ends were supported by two metal supports, with the defect region facing downwards. The force was then perpendicularly applied to the longitudinal axis of the bone by a cylindrical rush until the moment of fracture. The force applied and the indentator displacement were monitored and recorded using the equipment’s own software. From the force–displacement curve, fracture energy (J) (ability to absorb energy to breakage), elastic deformation energy (J) (ability to absorb and return energy without apparent deformation) and maximum load (N) (maximum load that the material can support) were obtained [23].

#### 2.8.7. Statistical Analysis

Initially, the variable distribution was tested using the Shapiro–Wilk’s normality test. For variables that exhibited normal distribution (BV/TV%; OV/TV%; N.Ob/T. Ar mm^2^; Ob.S/BS%; Fb.V/TV%; Bm.V/TV%; fracture energy (J); elastic deformation energy (J); maximum load (N)), comparisons among groups were carried out by analysis of variance (ANOVA), followed by Tukey post hoc. The Mann–Whitney test was used for variables not exhibiting normal distribution (characterization tests: pH and mass degradation). The statistical program used was GraphPad Prism version 7.0 and the adopted significance level was 5% (*p* ≤ 0.05).

## 3. Results

### 3.1. Characterization of Scaffolds

#### 3.1.1. SEM 

The qualitative analysis of the surface morphology of scaffolds showed the presence of silica spicules and pores in both materials (scaffolds of *Dragmacidon reticulatum* and *Amphimedon viridis* marine sponges) (Figure 1). *Dragmacidon reticulatum* sponge scaffolds were structurally more porous than *Amphimedon viridis* sponge scaffolds. Additionally, it was observed that the scaffolds of the *Dragmacidon reticulatum* species show a greater degradation after incubation when compared to the *Amphimedon viridis* species (Figure 3). 

#### 3.1.2. FTIR

The scaffolds manufactured from the two species of marine sponges studied (*Dragmacidon reticulatum* and *Amphimedon viridis*) exhibited the same functional groups: at 3460 cm^−^^1^, a broad peak consistent with the intermolecular OH bond; weak peaks at 2950 cm^−^^1^ and 2875 cm^−^^1^ for asymmetric and symmetrical -CH2- (OH-linked), respectively; weak signal between 800 cm^−^^1^ and 960 cm^−^^1^ of rotary motion -CH2-; presence of medium to strong signal between 1260 cm^−^^1^ and 1440 cm^−^^1^ being positive for primary alcohol; average signal at 770 cm^−^^1^ of the silicon–carbon bond; SiOH peaks at 3700 to 3200 cm^−^^1^, 1030 cm^−^^1^ and 890 cm^−^^1^; presence of signals at 1650 cm^−^^1^ and 1535 cm^−^^1^ consistent with amide I and amide II functions, respectively. However, from the seventh day of incubation, there was loss of the characteristic point of amide I, amide II and primary alcohol (CH2OH)—related to organic matter—only in *Dragmacidon reticulatum* species, as shown in Figure 4.

#### 3.1.3. XRD

XRD analysis of the samples revealed that the two species of marine sponges studied are composed of amorphous content, as shown in Appendix A.

#### 3.1.4. EDS

In the EDS analysis, the scaffolds manufactured from the marine sponge species *Dragmacidon reticulatum* and *Amphimedon viridis* showed the same chemical composition as shown in Appendix A, and the three most proportionally present chemical elements were carbon, oxygen and silicon, which together represent more than 80% of the sample.

#### 3.1.5. pH and Mass Degradation

The results of the pH measurements during the incubation period are shown in Figure 5A. The scaffolds of the *Dragmacidon reticulatum* species showed an initial drop in pH in the periods referring to the 1st and 3rd days of immersion in PBS and subsequent increase from the 14th day, reaching pH similar to the initial value (pH = 7.6) at day 21. The pH of the medium incubated with scaffolds from the species *Amphimedon viridis* did not show significant variation during the analysis period, remaining practically stable from day 1 to 21. A statistical difference was observed in the comparison between the two species, in the first (*p* = 0.0158), third (*p* < 0.0001) and seventh (*p* = 0.0037) days of incubation.

Figure 5B shows the results of the degradation assays of the scaffolds’ mass after immersion in PBS at different periods. An initial loss of mass in both groups (day 1) was observed, being more pronounced in the species *Dragmacidon reticulatum* when compared to the species *Amphimedon viridis*. After the third day, the values remained stable. Statistical difference was verified between species in the first (*p* = 0.0003), third (*p* = 0.0286), seventh (*p* < 0.0001), fourteenth (*p* = 0.0002) and twenty-first (*p* = 0.0027) periods.

#### 3.1.6. Porosity

The results of the porosity tests (Figure 6) demonstrate an average of 84 ± 5% porosity for *Dragmacidon reticulatum* scaffolds and 90 ± 2% for *Amphimedon viridis*. No statistical differences were found between *Dragmacidon reticulatum* and *Amphimedon viridis* scaffolds.

### 3.2. In Vivo Tests

#### 3.2.1. Qualitative Histological Analysis

An overview of representative histological sections for all the experimental groups is shown in Figure 7. For CG, bone formation was observed at the border of the entire defect, with osteoid areas around the newly formed bone. The DR, when compared to AV, presented new bone tissue points and a greater presence of osteoid tissue, mainly around the spicules of silica and newly formed bone tissue. In AV, a fibrous capsule was formed around the lesion area where the scaffold was implanted. Neoformed bone tissue was absent in this group and the areas of osteoid tissue were smaller, with only poorly isolated portions inside the defect.

#### 3.2.2. Histomorphometric Analysis 

Figure 8 shows the mean and standard deviation (SD) for the quantitative histomorphometric variables: BV/TV (%), OV/TV (%), N.Ob/T. Ar (mm^2^), Ob.S/BS (%), Bm.V/TV (%) and Fb.V/TV (%).

For the BV/TV parameter, it was verified that the bone volume that was formed 15 days after the surgical procedure was proportionally higher in the control group compared to the other two groups that included scaffold implants. On average, the percentage of bone volume in the analyzed area of the defect was 32.6 ± 8.5% for CG, 0.05 ± 0.04% for DR and 0.00% for AV. Statistical differences were found between the groups: CG vs. DR (*p* < 0.0001) and CG vs. AV (*p* < 0.0001), in the ANOVA/Tukey test, as shown in Figure 8A.

On the other hand, when evaluating the percentage of osteoid in the analyzed area of the defect (OV/TV%), it was observed that the formation of this tissue was significantly superior in the group with scaffold implantation of the marine sponge *Dragmacidon reticulatum* (DR; 8.3 ± 2.6%) in comparison to the control group (CG, 4.3 ± 1.9%) and *Amphimedon viridis* (AV, 0.7 ± 0.3%). A statistically significant difference was observed in the comparisons among all groups: CG vs. DR (*p* = 0.0042), CG vs. AV (0.0077), DR vs. AV (*p* < 0.0001) (Figure 8B).

Additionally, in the evaluation of the parameter referring to the percentage of biomaterial in the analyzed tissue (Bm.V/TV%), a significant statistical difference was observed between the groups CG vs. DR and CG vs. AV, where the CG 0%, DR 0.03 ± 0.01% and AV 0.04 ± 0.01%. Thus, there was no difference between the two types of scaffolds implanted (Figure 8C).

The number of osteoblasts per unit area of tissue analyzed (N.Ob/T.Ar mm^2^) of 132.0 ± 65.3 mm^2^ for the CG, 56.7 ± 11.3 mm^2^ for the DR and 13.2 ± 5.0 mm^2^ for AV was observed. There were statistical differences between the CG vs. DR and CG vs. AV groups (Figure 8D). 

On the other hand, the osteoblastic surface as a percentage of the bone surface (Ob. S/BS%) was 8.4 ± 5.6% for the CG, 17.7 ± 14.6% for DR and 0.0% for AV. A statistical difference was observed between the DR vs. AV (*) groups (Figure 8E).

In the evaluation of parameter Fb.V/TV (%), regarding the percentage of fibrous tissue in the analyzed tissue, statistical differences were observed between the CG vs. AV (*p* = 0.0068) and DR vs. AV (*p* = 0.0022) groups, as fibrosis was only verified in AV (19.9 ± 17.1%) (Figure 8F).

#### 3.2.3. Biomechanical Analysis

In the evaluation of the fracture energy (J), the respective mean values of 0.04 ± 0.02 J, 0.03 ± 0.02 J and 0.04 ± 0.02 J were observed for CG, DR and AV. The evaluation of the variable elastic deformation energy (J) showed the mean values 0.02 ± 0.01 J, 0.02 ± 0.01 J and 0.03 ± 0.01 J for the CG, DR and AV groups, respectively. Finally, in the evaluation of the maximum load (N), an average value of 49.2 ± 5.6 N was observed for CG, 47.6 ± 9.8 N for DR and 46.7 ± 10.9 N for AV. No statistical differences were observed among groups (*p* > 0.005) for any of the parameters (Figure 9).

## 4. Discussion

This study aimed to evaluate the physical-chemical properties of scaffolds manufactured from two species of marine sponges of the Demospongiae class and to investigate the in vivo biological response of these natural biomaterials once implanted in bone defects in rat tibia. The hypothesis was that, since different species could be physically and morphologically different, the osteogenic potential of these scaffolds would also be distinct. The main results showed that scaffolds of *Dragmacidon reticulatum* presented better performance in bone repair when compared to scaffolds of the species *Amphimedon viridis*.

Initially, SEM micrographs showed that the scaffold structures of both species are constituted by a porous material, with interconnected pores and presence of silica spicules, degrading after immersion in PBS. These characteristics are very important, since previous studies prove that the presence of interconnected pores and silica spicules are important characteristics in the scaffolds destined to favor the process of bone repair, as well as the gradual degradation of the scaffold allowing the replacement of the material by bone tissue [24,25]. Interestingly, scaffolds from both species exhibited a great porosity (84 ± 5% for DR and 90 ± 2% for AV), which would be adequate for their use as bone substitutes since the scientific literature establishes a minimum of 80% for an ideal scaffold [26]. In fact, higher porosity is expected to be reflected as an increase in bone formation, as observed in several studies involving different biomaterials, used as raw material for scaffolds’ manufacture [27]. In the study of Nandi et al. [15], SEM analysis also showed that the *Biemna fortis* skeleton has a collagenous fibrous network with highly networked porosity in the size range of 10–220 µm. However, the fibrous material was burnt at 725 °C and the fired material was mixed with naphthalene to induce porosity. After naphthane removal, apparent porosity was determined to be about 52%, which is substantially lower than the porosity found for *Dragmacidon reticulatum* and *Amphimedon viridis* scaffolds used in the present study (84 ± 5% for DR and 90 ± 2% for AV). This can be explained not only by the species-related difference, but also by the manufacturing method of the scaffolds themselves.

In the FTIR analysis, the scaffolds of both species had the same functional groups, with organic amide functions at 1650 cm^−^^1^ and 1535 cm^−^^1^, a primary alcohol at 2950 cm^−^^1^ and 2875 cm^−^^1^, inorganic silicon–carbon bonds at 1250 cm^−^^1^ and 770 cm^−^^1^ and silicon-hydroxyl at 3700 cm^−^^1^, 1030 cm^−^^1^ and 890 cm^−^^1^ [28]. It was observed that from the seventh day of incubation, for the *Dragmacidon reticulatum* species, there was a reduction in the points referring to the amide and primary alcohol groups, characteristic of oxidation of the organic matter, evidencing the degradation of the sample during the incubation time. This finding confirms our previous results of electron microscopy, which indicated a greater presence of pores and greater degradation in the scaffolds of this same species. For the species *Amphimedon viridis* the points remained stable during the incubation period evaluated. This difference between the species studied also suggests a better potential of the species *Dragmacidon reticulatum* when compared to the species *Amphimedon viridis*, since the study of Kido et al. [25] has shown that, in addition to porosity, proper degradation of the scaffold is essential for the bone repair process to occur. It is known that tissue formation depends on space to occur, and that the presence of a biomaterial may be a barrier to the growth of new tissue, especially if the implanted biomaterial does not have adequate porosity or a degradation rate compatible with the rate of bone formation. In the study of Nandi et al. [18], FTIR spectra revealed the presence of Si-O-Si groups at about 800 cm^−^^1^ and massive carbonate groups at about 1500 cm^−^^1^, which is a sign that the freeze-dried *Biemna fortis* material has a high organic content.

The XRD revealed that the scaffolds of the two species of marine sponges studied are composed of amorphous organic content, which coincides with the findings of Nandi et al. [18], who also characterized the marine sponge (*Biemna fortis*) belonging to the same class of species of this study (class Demospongiae). In addition, Schröder et al. [29], when studying the structure, biochemical composition and formation mechanism of biosilica produced by living organisms (marine sponges, diatoms and higher plants), reported it to be amorphous silica without evidence of crystalline silica. Gabbai-Armelin et al. [12] also found the mostly amorphous nature of biosilica extracted from *Dragmacidon reticulatum* species, with only a few crystalline peaks being observed. Therefore, these studies agree with the amorphous nature of sponges and their components, and the implication of this finding for bone tissue engineering would be the greatest biodegradation of these natural biomaterials once implanted, considering that the crystalline arrangement would imply greater stability due to the structural configuration of their monomers [3]. This may be beneficial for the replacement of the implant by the natural tissue, although potentially promoting, on the other hand, a significant decrease in the mechanical properties of the biomaterial [3].

For the EDS analysis, the scaffolds of both species had the same chemical composition, the three main components being carbon, oxygen and silicon, and a lower percentage of aluminum, calcium, chlorine, iron, potassium, magnesium, sodium, phosphorus and sulfur. These findings are in line with those obtained in the Sandford [30] study, which carried out a physical-chemical analysis of the siliceous skeletons of six species of marine sponges of two groups (Demospongiae and Hexactinellida), which concluded that the average chemical composition of the spicules of both groups was 85% silica (SiO_2_) and presented small amounts of the other elements mentioned earlier in this study. Additionally, Schröder et al. [29] reinforce that silica spicules, besides silicon and oxygen, present trace amounts of various other elements (mainly aluminum, calcium, chlorine, iron, potassium, sodium and sulfur), where approximately 75% of the body mass of these animals is silica. Interestingly, marine organisms process approximately seven gigatons of silicon to make their silica skeletons, an interesting feature for regenerative medicine in view of the osteoinductive potential (i.e., ability to stimulate differentiation of precursor cells in functional osteoblasts) of this component [31].

For the analysis of pH and mass degradation, a more marked variation of the values was observed, in the sense of an acidification of the medium with the loss of the material, during the first week of immersion of the scaffolds in PBS, reaching stability in the subsequent periods. It is interesting to note that only the DR scaffolds showed the initial drop in pH in the periods referring to the 1st and 3rd days of immersion in PBS and following. This could be explained by the fact that the loss of the scaffolds’ mass after immersion in PBS was remarkably greater in DR according to the degradation assays, with the products of this lixiviation most likely being responsible for the pH alterations. Further, this acidification would be interesting in the context of bone healing since Hazehara-Kunitomo et al. [32], who were able to measure the pH in vivo during the healing of a bone fracture, showed that the pH decreased to 6.8 during the inflammatory period (initial 2 days) and that this short-term acidification could help stem cell differentiation towards bone-forming osteoblasts [33].

In the histological analysis of the present study, bone formation was observed around the border of the whole defect in the CG, and histomorphometry (parameter BV/TV%) revealed that the percentage of bone formed was higher in this group than in the other groups that used marine sponge grafts (DR and AV). This can be explained by the adoption of a non-critical bone defect model that allows spontaneous bone repair with complete closure of the defect. In addition, other studies have shown that, initially, the presence of the biomaterial at the lesion site constitutes a barrier to bone growth, since the biomaterial needs to be degraded for the subsequent occurrence of bone tissue formation [34,35,36,37,38,39]. In the study of Nandi et al. [18], scaffolds prepared from *Biemna fortis* sponges (belonging to the same class of sponges of the present study—Demospongiae) were implanted into non-critical femoral bone defects in rabbits and tracked for up to 90 days. No quantifications were performed, but radiological, histological and scanning electron microscopy have demonstrated that bare sponge scaffolds (without any growth factor loading) promoted better osseous tissue formation, with invasion of this new bone across the porous scaffold’s matrix, in comparison to bone defects without any implant, which mostly showed soft tissue formation, with the defect gap still present even after 3 months.

In addition to mineralized bone, the formation of osteoid tissue was also evaluated in this study. It was present mostly underlying the newly formed bone tissue and around the sponge silica spicules, being much more evident in the scaffolds created from *Dragmacidon reticulatum* species. In fact, the analysis of the histomorphometric parameter OV/TV (%) showed a higher osteoid percentage in DR. Therefore, we can indicate more advanced bone repair in the CG due to the greater presence of mineralized bone tissue, followed by DR, with a more important presence of bone matrix that is not yet mineralized and, finally, AV, in which the neoformation of osseous and osteoid bone tissue was lower than the others. Nandi et al. [18] also assessed newly formed osteoid tissue in implanted *Biemna fortis* scaffolds through fluorochrome labeling—oxytetracycline. They showed a greater presence of newly formed bone in sponge scaffolds in comparison to control defects, which were moderately filled with osteoid tissue and where consolidation was under process.

It is important to note that the evidence discussed above is consistent with the cellular parameters analysis. Several in vitro studies have shown that the inorganic part of the sponge skeleton (biosilica) induces bone neoformation by attracting osteoprogenitor cells and stimulating their differentiation in osteoblasts [3,26,31,34]. In the present study, the number of osteoblasts per unit area of tissue analyzed (N.Ob/T.Ar/mm^2^) was higher in the CG compared to the other groups, which is compatible with the presence of a greater amount of neoformed bone tissue. Additionally, analyzing the osteoblastic surface (Ob.S/BS%), which refers to the percentage of bone surface covered by active osteoblasts, a statistical difference was found between the two groups with sponge implants, DR (17.7 ± 14.6%) and AV (0.0%), being superior in the *Dragmacidon reticulatum* species, which again indicates superior osteogenic properties in the scaffolds manufactured from this species. In the study of Nandi et al. [18], despite the absence of quantitative parameters, well-formed osteons inside sponge-filled defects exhibited osteoblastic and osteoclastic activities, whereas osteoclasts were prominently observed at the cortical region of non-filled control defects. The authors state that sponge scaffolds most likely serve as a bioactive stimulant for cell maturation, in agreement with other studies that have previously show that, at least in in vitro situations, osteoblast attachment, proliferation, migration and differentiation can be stimulated by marine sponge components [10,11,38].

However, in the present study, in addition to the lower bone and osteoid formation and the lower number of osteoblasts in AV, a fibrous capsule formation was observed around the implant area only in this group, which may indicate an attempt by the organism to isolate the biomaterial, i.e., a rejection of *Amphimedon viridis* scaffolds [39]. Urabayashi [19] has demonstrated, through the in vitro chemical analysis of the crude sponge extract of this species, a cytotoxic effect in human cell lines from retinal pigment epithelium and breast carcinoma, as well as a hemolytic action in rat cells, which may explain the results of the present study, since severe local and systemic inflammatory and cytotoxic responses caused by the implants may result in delayed or non-healing of the bone [25]. Therefore, despite previous research demonstrating the osteogenic potential of marine sponges in terms of both bone formation and cell stimulation, the current study shows that the benefits may also depend on the species of sponge studied and that the integration of the material with the original tissue can be significantly inferior or even harmful depending on the species, as shown here.

As the last histomorphometric parameter, the percentage of biomaterial inside the defect (Bm.V/TV%) was not different between DR and AV, although the characterization assays indicated a more important degradation in scaffolds manufactured from the *Dragmacidon reticulatum* species. Therefore, studies with longer experimental periods are necessary in order to allow a longer time for the biodegradation of the implanted materials and for the concomitant process of bone consolidation. In the study of Nandi et al. [18], evidence of sponge scaffold degradation was found radiologically, mostly from day 60 when the scaffold altered its shape from cylindrical to oval, with the edges of the filling material reducing its size, a sign that newly formed bone tissue was beginning to replace the sponge material. Moreover, the authors found that the enrichment of the scaffolds with the IGF-1 and BMP-2 growth factors promoted bone formation, with the presence of neoformed bone tissue and active osteoblastic cells throughout the defect region, as well as increased biodegradation of the implanted biomaterial.

Finally, for all the parameters evaluated in the biomechanical test, no statistical differences were found among the groups, although the CG had a higher bone formation than the others. This may be due to the presence of the biomaterial at the site of the defect in the two other groups, which could imply a greater resistance to fractures. In addition, interestingly, it may indicate a good integration of the implanted materials with the pre-existing living tissue, a hypothesis also reported in the study by Granito et al. [3] involving bone defects in tibiae of rats filled or not with bioactive synthetic materials.

Together, the characterization analyses and the in vivo study with the scaffolds manufactured from the species *Dragmacidon reticulatum* and *Amphimedon viridis* confirmed our hypothesis that different species, because they exhibit different chemical and structural properties, also have different osteogenic properties. Here, the *Dragmacidon reticulatum* species proved to be a more interesting option as a scaffold in bone tissue engineering and, therefore, the so-called good osteogenic potential is species-dependent, as demonstrated herein in a pioneering way.

In a pioneering way, this study compared two abundant species on the Brazilian coast and performed an extensive quantitative analysis of the bone repair process after in vivo implantation of these natural biomaterials. However, there are some limitations that still need to be addressed herein, such as the use of a non-critical bone defect model, which was chosen to allow biomechanical evaluations on a weight-bearing bone, and the short treatment period after scaffold implantion. Therefore, the critical-sized calvarial defect could be the next step of this study, together with longer follow-up periods and the inclusion of other experimental groups for the evaluation of scaffolds manufactured from marine sponges belonging to other species and even other families. Actually, a larger number of different species and families could have their potential evaluated if in silico methods had been employed. Faster outcomes and reduced costs are some of the benefits that computer simulations could bring for the prediction of the implant failure values, as in the study of Putra et al. [40]. Finally, another limitation of this study is inherent to the use of a natural biomaterial itself. Structural and morphological variations are expected among specimens collected from the environment. Subsequently, biomaterial availability would be an additional challenge, especially considering the clinical application as the final purpose of this study. Both obstacles could be at least partially overcome with the production of sponge biomass by in situ aquaculture. Mariculture could possibly satisfy the biomaterial demand without compromising natural sponge beds, constituting, therefore, a sustainable approach for biomaterial production [41].

In addition, the biological performance of scaffolds should also be assessed in other experimental animal models, as well as including the evaluation of bone repair in clinical situations in which it is compromised, such as in diabetes, in order to validate the use of these natural biomaterials as promising therapeutic alternatives.

## 5. Conclusions

It can be concluded that scaffolds manufactured from the marine sponge species *Dragmacidon reticulatum*, when compared to *Amphimedon viridis* species, were more effective as bone substitutes, since their structure was more porous and biodegradable, with a compatible increase in osteoid tissue formation and presence of osteoblastic cells, despite the absence of fibrous tissue formation in the studied period. Thus, sponges of this species may constitute a promising alternative source of biomaterials for use in bone tissue engineering. Further research is still needed for a better understanding of these marine sponge scaffolds. A critical bone defect model in rats should be the next step of this study, as well as other experimental models to evaluate the bone repair when it is compromised, such as an osteoporosis model. In addition, another crucial perspective for the future would be the improvement of mariculture techniques in substitution of sponge collection in the environment. Thereby, the sustainable obtainment of sufficient amounts of biomaterials would be achievable in view of their envisaged therapeutic application.

## Figures and Tables

**Figure 1 jfb-14-00122-f001:**
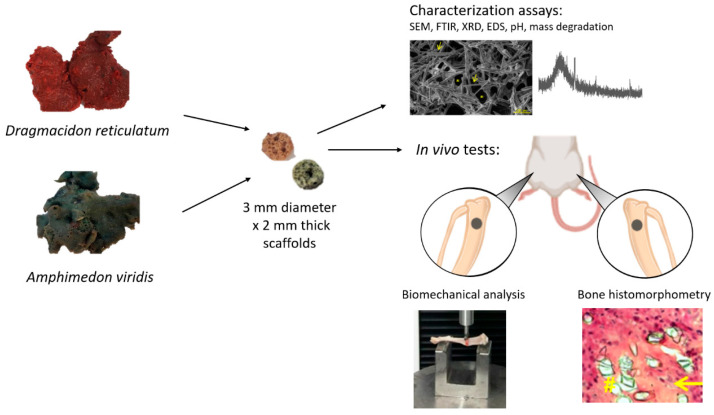
Summary of research methodology showing samples of marine sponges of the species *Dragmacidon reticulatum* and *Amphimedon viridis*, scaffolds manufactured from these sponges and the experiments performed.

**Figure 2 jfb-14-00122-f002:**
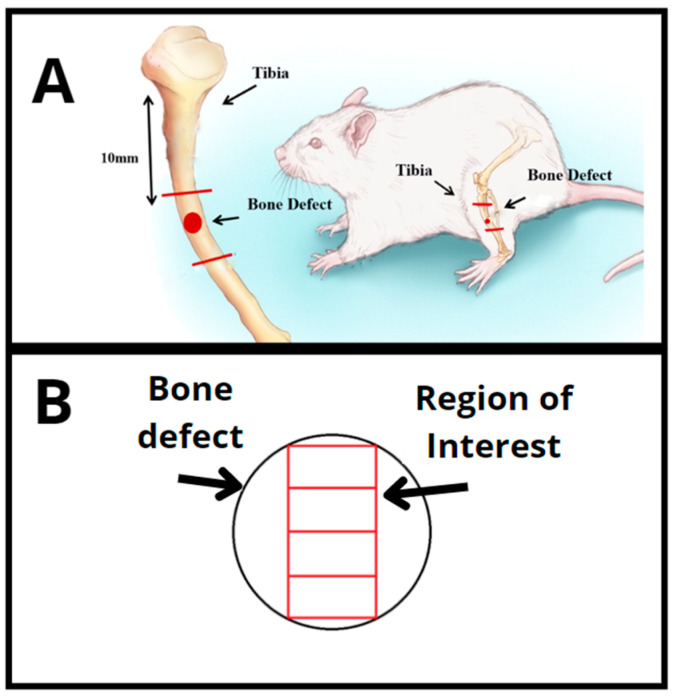
(**A**) Scheme showing the procedure for creating the tibial bone defect in rats. (**B**) Total region of interest analyzed for the histomorphometric parameters.

**Figure 3 jfb-14-00122-f003:**
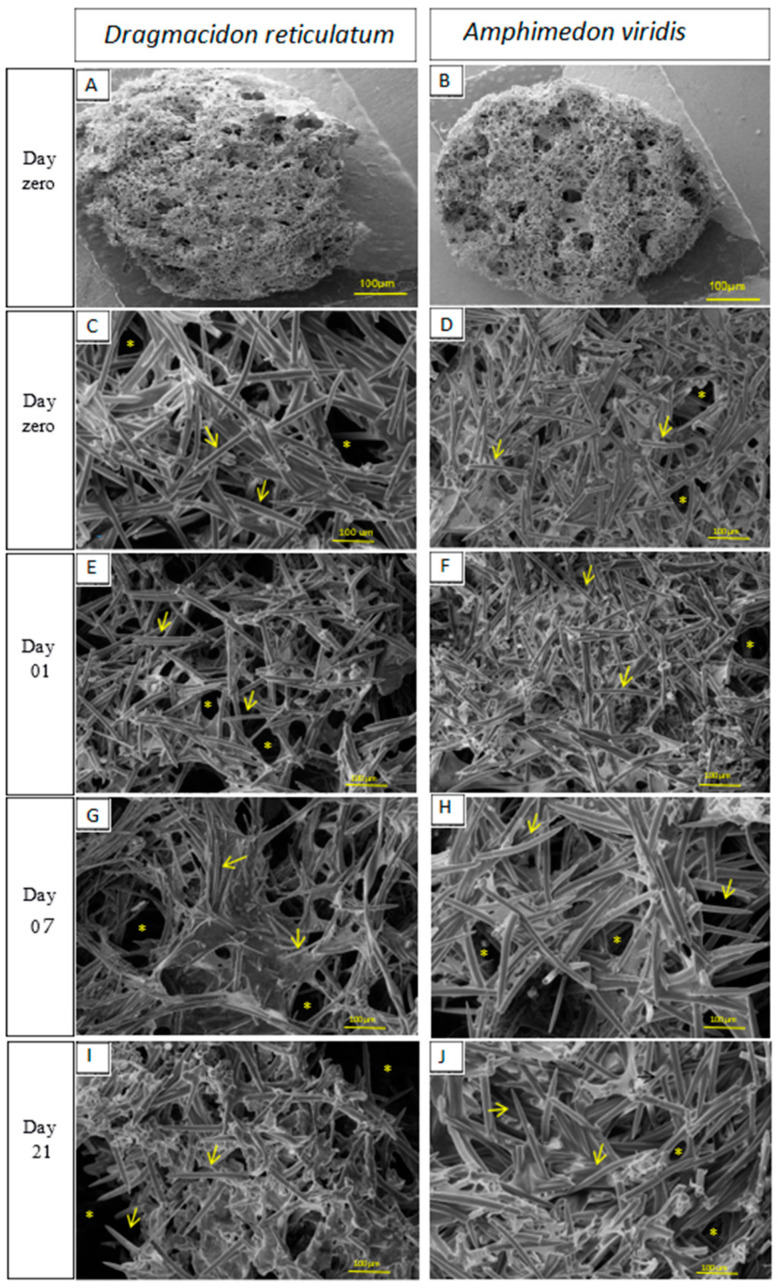
SEM micrographs showing scaffold morphology and its surface degradation in the periods before incubation (day zero (**A**–**D**)) and after incubation in SBF (days 1(**E**,**F**), 7(**G**,**H**) and 21(**I**,**J**))—increase of 500×. The signs indicate the presence of silica spicules (→) and pores (*). (*n =* 5).

**Figure 4 jfb-14-00122-f004:**
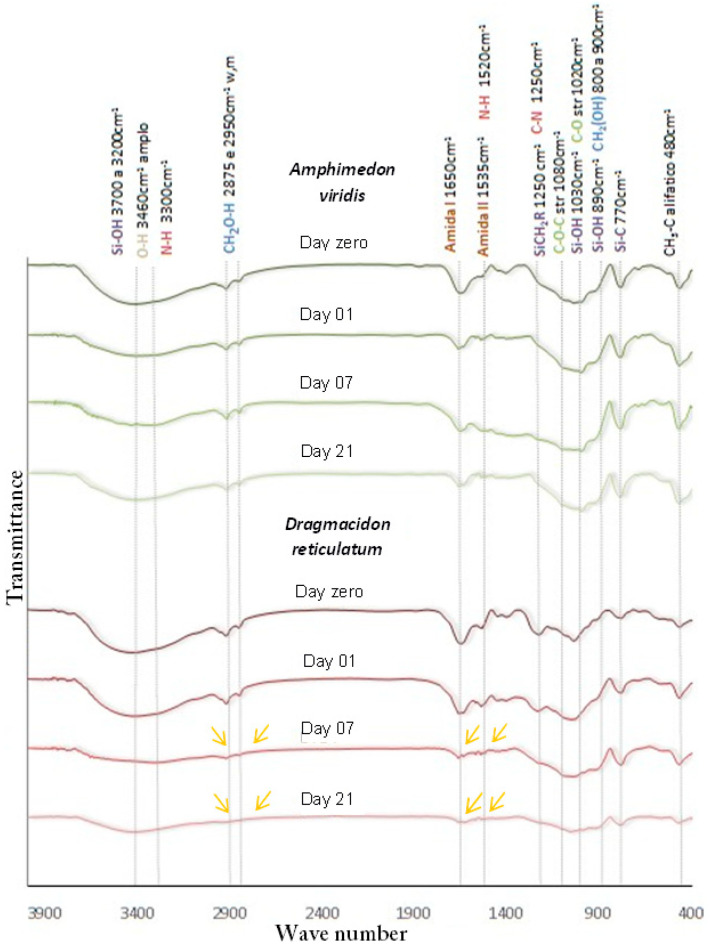
FTIR spectra of the scaffolds manufactured from the marine sponge species *Dragmacidon reticulatum* and *Amphimedon viridis*, before (day zero) and after incubation in SBF (day 1, 7 and 21). The yellow arrows show the loss of organic matter after incubation for *Dragmacidon reticulatum (n* = 3).

**Figure 5 jfb-14-00122-f005:**
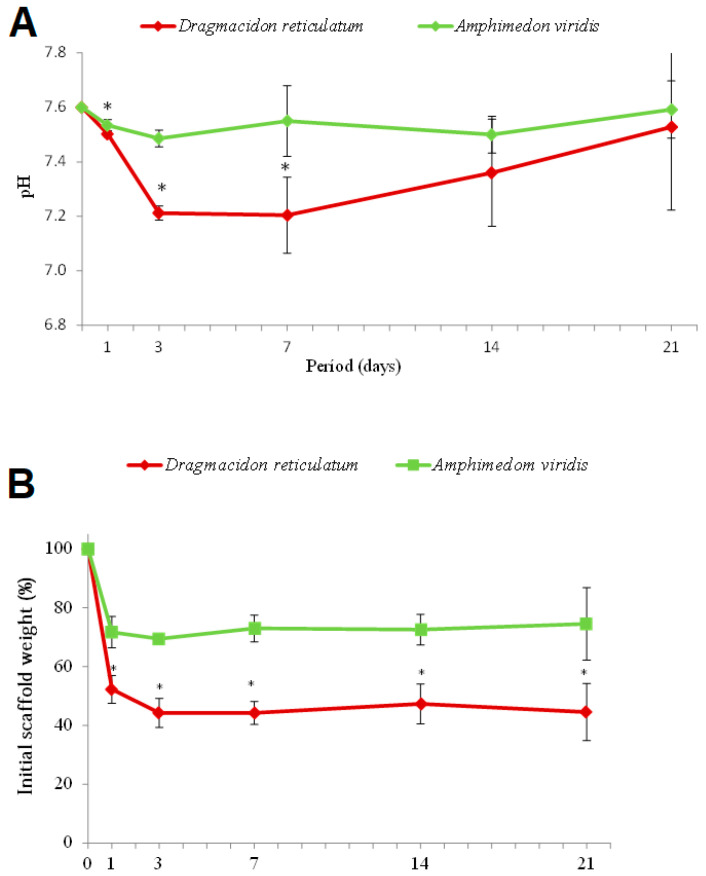
(**A**) pH values after incubation of the scaffolds in PBS. Means ± SD * *p* < 0.05 vs. *Dragmacidon reticulatum* (Mann–Whitney test). (**B**) Mass degradation of the scaffolds after incubation in PBS. Means ± SD * *p* < 0.05 vs. *Dragmacidon reticulatum* (Mann–Whitney test) (*n* = 5).

**Figure 6 jfb-14-00122-f006:**
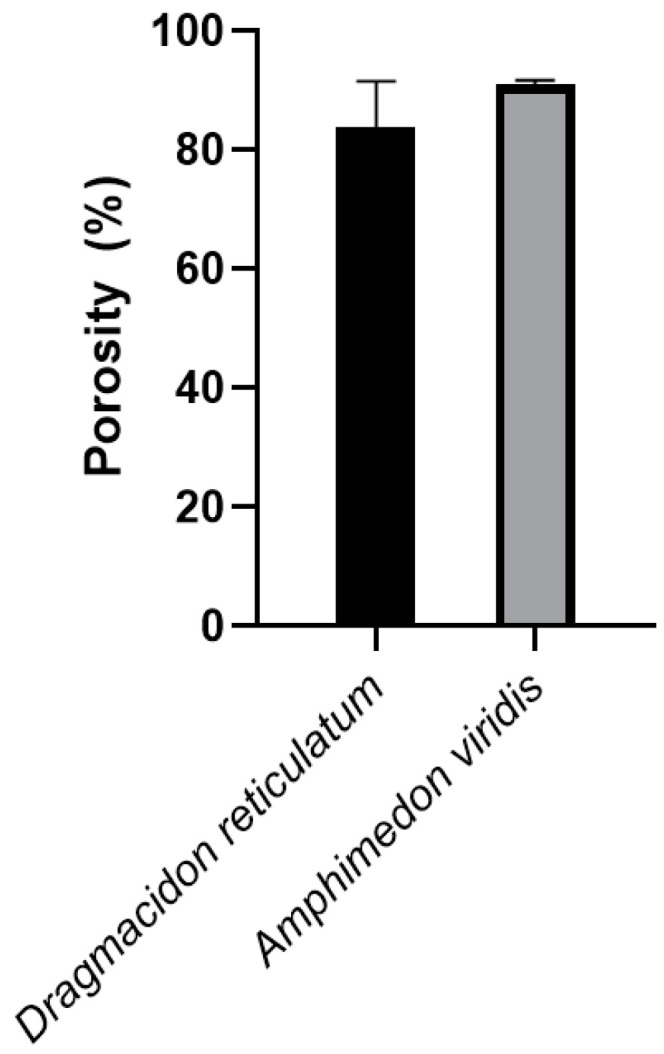
Porosity test of the sponges’ scaffolds. No difference was found between *Dragmacidon reticulatum* scaffolds and *Amphimedon viridis* scaffolds (*n* = 5).

**Figure 7 jfb-14-00122-f007:**
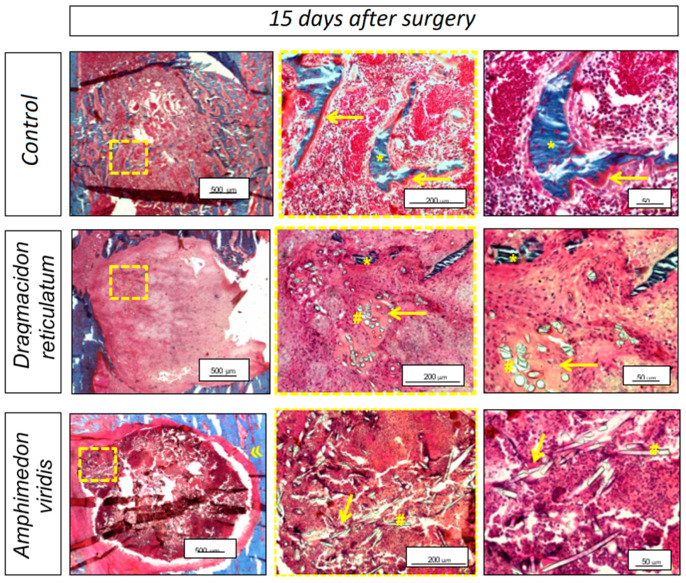
Bone defect photomicrography 15 days after the surgical procedure for the untreated experimental group (CG), for the group in which scaffolds of the marine sponge *Dragmacidon reticulatum* were implanted in the bone defects (DR) or for the group in which implanted scaffold belonged to the *Amphimedon viridis* species (AV). Neoformed bone tissue (*); osteoid (→); silica spike (#); and fibrous capsule (»). Bar scale = 500 μm (2.5× image), bar scale = 200 μm (20× image) and bar scale = 50 μm (40× image). Used stain: Goldner’s tri-chromium (Merck).

**Figure 8 jfb-14-00122-f008:**
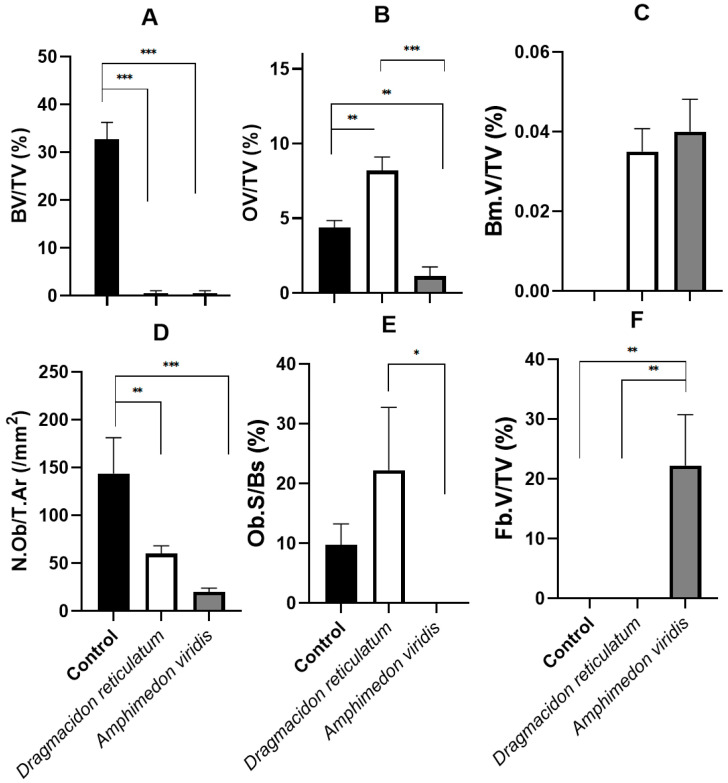
Bone histomorphometric analysis of bone defects 15 days after the surgical procedure for the experimental groups: no treatment (control—CG); submitted to the scaffold implant of the marine sponge *Dragmacidon reticulatum* (DR); submitted to the implant of the species *Amphimedon viridis* (AV). (**A**) Bone volume as percentage of tissue volume (BV/TV%), means ± SD *** *p* < 0.0001 (Anova/Tukey test). (**B**) Osteoid volume as percentage of tissue volume (OV/TV%), means ± SD ** *p* < 0.01; *** *p* < 0.0001 (Anova/Tukey test). (**C**) Biomaterial volume as percentage of tissue volume (Bm.V/TV%), means± SD *p* > 0.05 (Mann–Whitney test). (**D**) Number of osteoblasts per unit tissue area analyzed (N.Ob/T.Ar (mm^2^)), means ± SD ** *p* = 0.0039; *** *p* < 0.0001 (Anova/Tukey test). (**E**) Osteoblastic surface as a percentage of the bone surface (Ob.S/Bs%), means ± SD * *p* = 0.0237 (Anova/Tukey test). (**F**) Percentage of fibrous tissue in the analyzed tissue (Fb.V/TV%), means ± SD ** *p* = 0.0068; *p* = 0.0022 (Anova/Tukey test).

**Figure 9 jfb-14-00122-f009:**
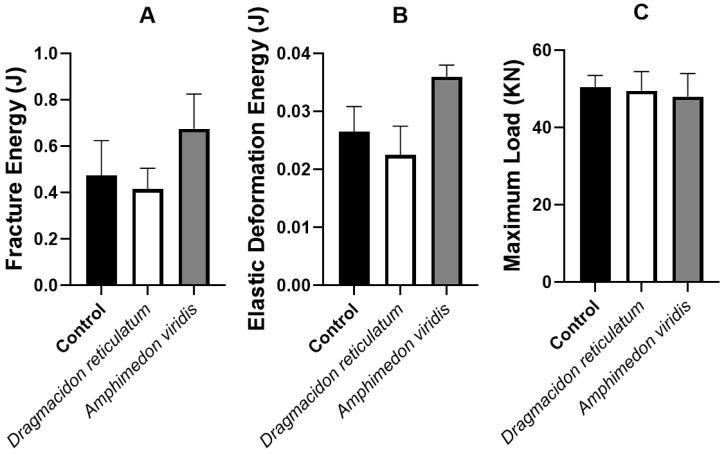
Mean and standard deviation of the variables of the biomechanical test: (**A**) fracture energy (J), (**B**) elastic deformation energy (J) and (**C**), and maximum load (N). Means ± SD *p* > 0.05. (Anova/Tukey test).

## Data Availability

The data presented in this study are available on request from the corresponding author.

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
