# Peer review of "Different Species of Marine Sponges Diverge in Osteogenic Potential When Therapeutically Applied as Natural Scaffolds for Bone Regeneration in Rats"

_jfb, 2023, doi:10.3390/jfb14030122_

Round 1

Reviewer 1 Report

Dear authors,

I have gonethrough the manuscript thoroughly and found the study to be interesting and biologically relevant. But The writing style needs to be improved along with the grammatical errors. I have some suggestions and comments that can enhance the readability and quality of the manuscript.

1. Abstract (Line no. 16-21): Please change the sentence because the meaning of the sentence is not relevant according to your hypothesis.

2. Line no.-47-48: Please correct the grammar of the sentence.

3. Line no. 72-76: Please re-frame the sentence for better understanding (sentence fragmentation required).

4. Line no. 83: Please delete word still. S"till" makes the meaning of the sentence different.

5. Line no. 304: Please mention the full form of AG and CG.

6. Line no. 388: Please reframe the sentence, because 80% of what is not understood properly.

7. Can you state the purpose of taking sponges of same family?

8. Please use either abbreviations or full forms of the species throughout the whole manuscript for better readability and formalization.

9. Please cite some recent references.

Reviewer 2 Report

Overall the manuscript is interesting, the novelty is not evident but the application of scaffolds prepared from sponges is. However, some recommendations should be taken into account:

1. The abstract lacks quantitative results, all are qualitative, which makes this item very descriptive and lacks depth.

2. The words in vivo, in vitro, in situ, in silico, are derived from Latin and should therefore be written in italics.

3. In the introduction, the low abundance of sponges for purposes such as those pursued by the present research is tangentially discussed, how to propose that sponges can be a promising or sustainable source for the production of these biomaterials?

4. In the materials and methods, the collection of the samples should be more detailed, how much was taken from each one of the sponges, how was their classification carried out, and, how were they kept until the preparation of the scaffolds.

5. What were the freeze-drying and sterilization conditions of the specimens?

6. The SEM analysis does not mention how many samples were analyzed? how many microphotographs were taken?

7. Correct on line 136, "37°C"

8. In the porosity evaluation, clarify how the scaffolds were dried before weighing and how they were maintained so that they did not absorb moisture from the environment?

9. Number the equations. 

10. Figure 4. n = ?

11. Figure 5. n = ?

12. Figure 7. n = ?

13. Figure 8. n = ?

14. The analysis of the results does not explain the decrease in the pH of the DR scaffolds.

15. Lines 473-4 talks about the number of osteoblasts per unit area, how were they counted if no staining was used, how many microphotographs were analyzed to give the result?

16. How can "active osteoblasts" be discriminated by morphology?

17. In vitro assays of cytotoxicity, migration and invasion of osteoblasts on the different scaffolds are recommended in order to determine their viability in their use.

18. It is recommended to redefine the keywords, as they are very general.

Reviewer 3 Report

The Article is devoted to an important field of searching the new therapeutic resources for the improvement of bone tissue defects. The study of natural biomaterials would help find a promising candidate which would be characterized by a combination of chemical composition, structural peculiarities, and biodegradation properties appropriate for bone-tissue-engineering applications.

The sponge species, which porous skeleton composed of spongin and biosilica, belonging to the one class were used in the research, which is very interesting because even non obvious differences in scaffolds physical-chemical characteristics might play a significant role in tissue regeneration process.

The intense study had been planned and carried out by the Authors. Different methods were applied during the research (SEM, FTIR, EDS, XRD, pH, mass degradation and porosity tests), the obtained results are quite informative and help understand better the properties of the material, in vivo tests show the specificity of bone tissue formation in contact with the substitutes.

The research stresses the importance of material degradation ability and character of organic matter loss, which both effect the process of bone tissue formation at the place of damage.

It would be interesting to know the mechanism which lies at the basis of the tissue reaction to the different materials, to have a possibility to predict the behavior of the organism response to scaffolds of various nature.

The results are logically presented, described in details, illustrated with histograms and images which make the presented data clear.

There is a question, why did the Authors decide not to use specific test for assessment of the inflammation process in the body after the implantation procedure (like C-reactive protein test)? It will be very informative.

And it would be better to start with the study of cytotoxicity of the biomaterials using cell cultures assays, before the in vivo experiments were planned. Such type of an assay helps to understand if the material would be biocompatible, which will allow to make the changes in the plan of the research if needed.

It is very interesting that Authors offer to conduct research with longer experimental periods. It also would be quite interesting to know the influence of studied biomaterials on the process of vascularization.

The following comments do not diminish the value of the Article

Line 4 The dot at the end of the title should be removed.

Line 95 The following phrase “Amphimedon viridis.” should be replaced with “Amphimedon viridis,”.

Line 99 What are the components of ‘simulated body fluid, SBF’? Would it be possible to characterize the fluid a bit more in details?

Line 123 It would be better to describe the following equipment ‘Philips, Cu-Ka, 45 kV, 30 mA’ in details.

Line 151 Porosity evaluation is not very much clear. Is that possible to make a description to the formula a bit detailed?

Lines 214, 215 It would be better to add a detailed characterization of the following equipment ‘Instron universal testing machine, model 4444’.

Line 244 A Figure and its caption should appear on the same page.

Line 287 A Figure and its caption should appear on the same page.

Line 298 Probably the equal font style should be used for titles.

Line 309 It would be better to describe the contrast enhancing technique that was used to obtain the specimens in the Figure 6 caption.

Line 322 As there is less amount of bone tissue at the place of defect in experimental groups probably it would be good to make the experiment for a longer time or carry out cytotoxicity tests in 2D cell cultures before the in vivo experiments.

Line 444 The names of the species ‘Spongia Agaricina’ and ‘Spongia Lamella’ should be italicized.

Line 513 The term ‘Desmonpongiae’ should be corrected.

Please check the References part and correct it according the examples presented at the website of the Journal.

Reviewer 4 Report

1.      The abstract should be broadened to give additional quantitative results.

2.      After line 23, please add the abstract's "take-home" message, the current form was insufficient.

3.      Reorder keywords based on alphabetical order.

4.      Describe the novelty of the article made by the author? From the results of my evaluation, it seems that many similar published works adequately explain what you have raised in the current manuscript. If there is something others really new in this manuscript, please highlight it more clearly in the introduction section.

5.      Previous study related needs to explain in the introduction section consisting of their work, their novelty, and their limitations to show the research gaps that intend to be filled in the present study.

6.      As the present study performs in vitro study, potential in silico study of scaffold needs to be explained since it bring several advantages such as faster results and lower cost. The suggested reverence published by MDPI should be adopted as follows: Level of Activity Changes Increases the Fatigue Life of the Porous Magnesium Scaffold, as Observed in Dynamic Immersion Tests, over Time. Sustainability 2023, 15, 823. https://doi.org/10.3390/su15010823

7.      Some materials adopted in scaffold needs to explained briefly in the introduction, such as metals and polymers. Suggested reference in comments number 6 would be adopted for this manner.

8.      To help the reader grasp the study's workflow more easily, the authors could include more visuals to the materials and methods section in the form of figures rather than sticking with the text that now predominates.

9.      It's also important to provide more particular information on tools, such as the manufacturer, the country, and the specification.

10.   The revised manuscript after peer review must provide detailed information on the error and tolerance of the experimental equipment utilized in this study. Due to the disparate outcomes of other researchers' subsequent studies, it would make for a valuable discussion.

11.   A comparative assessment with similar previous research is required.

12.   The authors need to improve the discussion in the present article become more comprehensive. The present form was insufficient.

13.   Please include the limitation of the present study, it is missing.

14.   Information to the conclusion by formatting it as a paragraph rather than the manner as point-by-point.

15.   Mention further research in the conclusion section.

16.   The authors should give additional references from the five-years back. MDPI reference is strongly recommended.

17.   English needed to be proofread by authors due to grammatical mistakes and English style.

18.   It is suggested to the authors for providing graphical abstract in the system after revision.

Round 2

Reviewer 4 Report

Reviewers greatly appreciate the efforts that have been made by the author to improve the quality of their articles after peer review. I reread the author's manuscript and further reviewed the changes made along with the responses from previous reviewers' comments. Unfortunately, the authors failed to make some of the substantial improvements they should have made making this article not of decent quality with biased, not cutting-edge updates on the research topic outlined. In addition, the author also failed to address the previous reviewer's comments, especially on comments number 4 (weak novel), 5 (unclear state of the art), 6 (not incorporated the literature), and 12 (poor explanation). Thank you very much for the opportunity to read the author's current work. 
